# Cancer-Associated Fibroblasts: Implications for Cancer Therapy

**DOI:** 10.3390/cancers13143526

**Published:** 2021-07-14

**Authors:** Ana Maia, Stefan Wiemann

**Affiliations:** German Cancer Research Center (DKFZ), Division of Molecular Genome Analysis, Im Neuenheimer Feld 580, 69120 Heidelberg, Germany

**Keywords:** cancer associated fibroblast (CAF), tumour microenvironment (TME), cell communication, signalling, therapy resistance

## Abstract

**Simple Summary:**

Cancer-associated fibroblasts are important players of the tumour microenvironment. They influence numerous processes during tumour development and progression, including the response of cancer cells to treatment. As a consequence, this cell type has emerged has a prominent target in anti-cancer therapy. In this review, we discuss the function and heterogeneity of fibroblasts as well as their role during treatment. Moreover, we describe how different therapies influence the phenotype of this cell type and the implications of these alterations. Finally, we provide a detailed overview of the current strategies employed in the targeting of fibroblasts as well as future perspectives. We believe that further dissection of the heterogeneity of fibroblasts and of their dynamics, not only during tumour progression, but also in the course of treatment is essential for successful targeting of this cell type and, consequently, for improving patient survival in cancer.

**Abstract:**

Tumour cells do not exist as an isolated entity. Instead, they are surrounded by and closely interact with cells of the environment they are emerged in. The tumour microenvironment (TME) is not static and several factors, including cancer cells and therapies, have been described to modulate several of its components. Fibroblasts are key elements of the TME with the capacity to influence tumour progression, invasion and response to therapy, which makes them attractive targets in cancer treatment. In this review, we focus on fibroblasts and their numerous roles in the TME with a special attention to recent findings describing their heterogeneity and role in therapy response. Furthermore, we explore how different therapies can impact these cells and their communication with cancer cells. Finally, we highlight potential strategies targeting this cell type that can be employed for improving patient outcome.

## 1. Introduction

The observation that tumour cells do not act as an isolated entity but, instead, interact with other cells in the human body was proposed in the mid-nineteen century by Rudolph Virchow who first established a link between inflammation and tumour development [1]. A further suggestion of the importance of the interactions of cancer cells and their microenvironment arose a few years later when Stephen Paget observed that disseminated tumour cells preferentially colonise certain organs and coined the famous ‘*Seed and soil hypothesis*’ [2]. Nowadays, a huge body of evidence has been obtained for the role of the TME in determining numerous aspects of tumour development, progression, metastasis development and therapeutic response [3]. The term TME (also named stroma) comprises a broad panoply, including all the components that surround the cancer cells, namely immune cells, vasculature, extracellular matrix (ECM) and fibroblasts, as well as their interactions [4].

The TME closely interacts with the malignant cells and can have either tumour restrictive or promoting effects. Moreover, these elements are not static and, as the tumour develops, the tumour cells modulate their environment and often highjack it to promote their growth and progression. An important and dominant component of the tumour stroma with broadly described pro-tumorigenic capacity is the cancer-associated fibroblast (CAF) [5]. CAFs are phenotypically different from normal quiescent fibroblasts and have been described to resemble fibroblasts in the context of wound healing due to their enhanced secretion of ECM components as well as other soluble factors such as cytokines and chemokines [6,7]. This is now known to be true for a subset of fibroblasts present in the tumour microenvironment, which can be identified based on their contractile characteristics and expression of certain proteins such as alpha smooth muscle actin (αSMA) [8,9]. However, recent advances in molecular technologies such as single-cell sequencing [10] have made the further dissection of stromal fibroblasts possible and has shown that this is not the whole story. Nevertheless, challenges remain in the characterization of fibroblasts and, consequently, their therapeutic targeting.

In this review, we summarise recent findings on the heterogeneity of CAFs and their implications for tumour biology with an emphasis on the roles of the different subpopulations in therapy response. Moreover, we highlight strategies for rational targeting of fibroblasts in order to improve therapy in the field of cancer.

## 2. Fibroblasts

Fibroblasts were first described in the 19th century by Virchow [11]. These mesenchymal-derived cells with a spindle-like appearance are the main cellular component of connective tissue and are responsible for the synthesis of the majority of the constituents of the fibrillar ECM, including collagens, elastin and fibronectin, which allows them to regulate the morphology of tissues [12]. In wound-healing, the role of these cells is very well established and the morphological and phenotypic changes that fibroblasts undergo have been widely described [13,14,15]. Even though the first description of these cells was almost two centuries ago, fibroblasts are often still identified based on their morphology, tissue location and lack of epithelial markers. The absence of specific markers that would allow the identification of this cell type has remained one of the biggest challenges in the study of fibroblasts [16]. Despite the lack of a universal marker, fibroblasts have thus far been identified by using a combination of commonly expressed mesenchymal markers and other proteins associated with certain activation states. The markers that are common, but not exclusive, to fibroblasts, regardless of their activation state, include vimentin [17], fibroblast-specific protein 1 (FSP-1, also known as S100A4) [18,19] and platelet-derived growth factor receptor α (PDGFRα) [20]. Several other markers have been shown to detect specific activation states of fibroblasts and, consequently, define subpopulations of fibroblasts. Among these, the most universally used are αSMA and fibroblast-activation protein α (FAPα) [21,22]. In addition to these, several markers specific for certain tissues have been identified. This high level of heterogeneity further complicates the connection between findings from one study to another, since uniformity in analysing this cell type is then missing. Novel advances in single-cell omics can overcome this major drawback and have already proven invaluable for advancing our knowledge in fibroblast biology.

The subpopulations of fibroblasts in healthy tissues are a reflection of different transcriptomic and epigenetic profiles that are dictated by the embryonic origin and architecture of the tissue as well as their microenvironment [23,24]. Using a variety of in vivo models, several studies aimed at dissecting fibroblast heterogeneity have identified numerous distinct subpopulations of fibroblasts in several tissues and diseases [24]. In humans, due to the higher complexity of sampling healthy tissues, studies have thus far extensively focused on skin fibroblasts, but also include other tissues. There, comparable to what was found in mouse models, several functionally distinct subpopulations have been shown to exist [25,26]. If fibroblasts in healthy tissues are already characterised by a high degree of heterogeneity, the picture in tumours is even more complex, not only due to possibly distinct origins [23], but also as a result of the dynamics and selective pressure that characterises tumour evolution [27]. Fibroblasts in the TME are exposed to a variety of factors, such as cytokines, chemokines and growth factors, whose composition can reprogram and dictate the phenotype of these cells. Several molecules have been described to activate and promote the pro-tumorigenic states of fibroblasts, including TGF-β, PDGF, FGF2, HGF, IL-1 and IL-6, among others [9]. In addition to reprogramming by secreted factors, fibroblasts have been shown to be activated by different stress sources, such as ROS and damaging therapy, and well as by alterations in the ECM [16].

A clear evidence of functional heterogeneity in fibroblasts present in the TME arose with the first trials to target this cell type in vivo. In pancreatic cancer, the depletion of αSMA-expressing fibroblasts from the TME accelerated tumour growth and resulted in an immunosuppressive TME [28,29]. Interestingly, the opposite effect was observed when CAFs expressing FAP were therapeutically targeted, with this resulting in an impairment of tumour growth [30,31]. As observed in a study by Özdemir and colleagues, this was also a result of alterations in the immune milieu of the tumours. Contrary to what happened with αSMA-expressing CAFs, the depletion of FAP-positive CAFs enhanced adaptive immunity and reduced immunosuppression in the TME [30,31,32]. This clearly illustrates that prior to any attempt to target these cells in the clinic, an improvement in our understanding in the biology of fibroblasts is essential. With this in mind, numerous groups have attempted to resolve this challenging problem and we will describe the main findings below. Although we have seen an explosion of studies whose goal is to unravel both the molecular and functional heterogeneity of CAFs, most of these have focused on investigating the composition of fibroblasts at different stages of tumour development, still leaving the picture incomplete. The impact of therapies in subpopulations of fibroblasts has been thus far largely overlooked and studies that investigate the evolution of fibroblasts alongside with other cell types, such as tumour and other TME cells, are absent. This could provide important insights in the biology of fibroblasts in therapy response and provide essential knowledge for the rational targeting of this cell type or of its interactions with tumour cells.

### 2.1. Fibroblast Heterogeneity

Prior to the development of single-cell sequencing, fibroblasts were often characterised based on the expression of pre-defined markers using flow cytometry analysis or immunohistochemistry. Despite the power of these techniques to provide single-cell and spatial resolution, an unbiased approach, in which the full transcriptome of fibroblasts was analysed without the pre-selection of certain markers, was missing. Bulk RNA-sequencing of fibroblast-enriched fractions based on negative selection provided a broad overview of gene expression in these cells but lacked the single-cell resolution. The development and establishment of single-cell RNA sequencing (scRNA-seq) provided the cornerstone for the study of tumour heterogeneity, including fibroblast heterogeneity. The development of microfluidic platforms that allowed a dramatic improvement in the power of analysis by increasing the number of sequenced cells [33], as well as the more recently proposed method of sequential indexing, which promises to exponentially increase the number of cells that can be sequenced at the single-cell resolution [34], are crucial developments to allow us to dissect the biology of fibroblasts in the context of cancer.

Based on the analysis of different tumour entities and using single-cell technologies, several fibroblast subpopulations have been identified and their roles in different aspects of the tumour start to be unravelled. The most well studied entities include breast [35,36,37] and pancreatic carcinomas [38,39,40,41,42], although studies in other tumour models are starting to appear [43,44,45,46,47]. Despite the context-dependent variability, two recurrent subpopulations that have now been identified in several tumour entities are inflammatory CAFs (iCAFs) and myofibroblasts (myCAFs). Very briefly, these subpopulations are functionally characterised by the secretion of inflammatory cytokines and cytoskeleton remodelling, respectively [38,40,46,48,49,50]. In addition to different transcriptional profiles, these subpopulations were shown to be located in distinct areas of the tumour, with myCAFs being found in closer proximity to malignant cells and iCAFs localising distally [38]. More importantly, these subpopulations have been shown to be functionally discrete and mutually exclusive [38,51]. Using single-cell sequencing, Öhlund et al. [38] first described these subpopulations in a model of pancreatic cancer. Their findings were further confirmed by other studies, where the transcriptional program of myCAFs was shown to be driven by TGF-β signalling and these cells were characterised by the expression of high levels of αSMA. On the other hand, the activation of the NF-κB pathway driven by IL-1β led to the acquisition of an iCAF phenotype. These cells were characterised by the secretion numerous inflammatory cytokines, including IL-6, CXCL1, CXCL12, among others [38,39,45]. These two subpopulations also seem to be regulated by sonic hedgehog (Shh) signalling, whose activation supported the myCAF phenotype [52]. Interestingly, iCAFs were shown to exclusively express HAS1 and HAS2, which are responsible for the synthesis of hyaluronan, a component of the ECM, which has been shown to work as a barrier in the treatment of pancreatic cancer [39]. In breast cancer, our group has shown that a subset of fibroblasts in the luminal subtype is characterised by the expression of NRG1, whose expression correlated with HAS2 expression [53]. Moreover, Biffi et al. have shown that the two fibroblast states, iCAF and myCAF, seem to be mutually exclusively, since exposure to TGF-β resulted in the downregulation of the IL-1β receptor (IL-1R1) and, consequently, led to the differentiation of fibroblasts into myCAFs [51].

The observation that these two fibroblast populations are recurrently found in the microenvironment of several tumour entities could arise from the fact that these are numerically dominant subpopulations. Nevertheless, this does not invalidate the existence of other subpopulations. In fact, several other subpopulations have been shown to co-exist, although these were more variable and different compositions have been identified according to the tumour entity studied. Moreover, it could be that these dominant subpopulations can still be further divided into further distinct functional clusters. In a recent study, Friedman et al. show three different clusters of inflammatory CAFs, each one being characterised by the expression of IL-6, CXCL1 and CXCL12. However, a real functional distinction between the three clusters is not shown in this study [36]. Furthermore, using flow cytometry, Costa et al. showed that fibroblasts in breast cancer could be divided into four populations (CAF-S1 to S4), according to the expression of commonly used markers such as FAPα, αSMA, FSP1, PDGFRβ and Caveolin, demonstrating an immunosuppressive role of CAF-S1 in the TME [37]. By analysing almost 20,000 cells from this CAF subpopulation using single-cell sequencing, Kieffer et al. could divide CAF-S1 into eight distinct clusters. They further show that fibroblasts from cluster zero and three were characterised by TGF-β signalling and high αSMA expression and that these cross-talked with T-cells in the TME, driving a regulatory phenotype in the T-cells (Treg) [54], which explains the immunosuppressive role initially proposed by the same group [37]. As described above, the targeting of all αSMA-positive cells and the depletion of these from the TME by Özdemir et al. led to the outgrowth of pancreatic tumours [28,29]. Although the study from Kieffer et al. focused on breast cancer, their findings clearly raise the question of whether the targeting of specific clusters of αSMA-positive fibroblasts could indeed be beneficial for patients to elicit an efficient T-cell mediated immunity against malignant cells. This highlights the importance of the advances seen in single-cell omics that allow us to improve the power of single-cell analyses and that will help us further dissect the complexity of fibroblasts in cancer.

Yet another interesting subpopulation that has also been recently identified in pancreatic cancer is termed an antigen-presenting CAF (apCAF). These are characterised by the expression of genes associated with MHC-class II antigen presentation [39], although they lack co-stimulatory ligands necessary to activate immune cells. Instead, MHC-II signalling in this subpopulation of fibroblasts seems to drive the expansion of regulatory T cells and, in this way, promote an immunosuppressive microenvironment [39]. Despite all these novel findings, further functional characterisation and validation of the importance of these different subpopulations in patients is still needed. Another layer of complexity arises when these subpopulations are analysed in the context of time, with studies showing alterations in the composition of these subpopulations between early and late tumours [36,40,42,45].

In the next section, we will describe the current knowledge on how fibroblasts affect the response of tumour cells to anti-cancer therapies and try to correlate the subpopulations of fibroblasts described above with these events. We mostly focus on interactions that result in a negative impact on therapy response and can, ultimately, result in therapeutic failure. Nevertheless, it is important to mention that not all interactions between CAFs and cancer cells drive tumour progression or resistance to therapy. Numerous studies have shown that CAFs can have anti-tumorigenic functions in the TME [55], strengthening the view that further dissection and understanding of the phenotypes and functions of fibroblasts is essential prior to their therapeutic targeting.

### 2.2. Fibroblast and Therapy Response

Cancer cells can exploit a variety of mechanisms to escape therapy-induced cell death or irreversible cell cycle arrest. These can be categorised into (1) drug-dependent mechanisms, (2) target-dependent mechanisms or (3) drug- and target-independent mechanisms. Very briefly, drug-dependent mechanisms include any alteration that interferes with the availability of the drug. Mutations in the target protein that interfere with the drug activity or alterations in signalling pathways that overcome target inhibition are some examples of target-dependent mechanisms. Finally, the last category includes the acquisition of new aberrations that switch the growth dependency of tumour cells to a different protein and/or pathway as well as alterations in apoptotic pathways [56]. The TME dictates the way cancer cells respond to therapy. Several studies investigating mechanisms of resistance nicely showed that in vivo resistant cancer cells lose their resistant phenotype when they are removed from their microenvironment and are grown alone in vitro [57,58]. Fibroblasts can influence the vast majority of the mechanisms mentioned above by either (1) supporting specific subpopulations of cancer cells, such as stem-like cancer cells [59,60,61,62,63], (2) rewiring the signalling networks in cancer cells [64,65,66,67] and (3) modulating the tumour microenvironment, namely immune cells [68]. Interestingly, spatial analysis has revealed that the close proximity of malignant cells to stromal fibroblasts is associated with an increased cycling capacity of cancer cells after therapy, hinting to a protective role of fibroblasts [69].

A large number of studies have shown that, through a variety of mechanisms, fibroblasts promote resistance to therapy due to their interplay with cancer stem cells (CSCs). This is especially true for the current standard of care therapies that often depend on the proliferative state of cancer cells, such as radiotherapy and chemotherapy. This is an indirect effect, since fibroblasts promote the expansion of the subset of cancer cells with stem-like properties that have been extensively described to be intrinsically more resistant to numerous therapies, but do not directly modulate the way cancer cells respond to the agent they are exposed to. A more direct mechanism through which fibroblasts can modulate the response of cancer cells to therapy is by activating alternative signalling mechanisms, such as MAPK and PI3K signalling, that provide pro-survival signals and, thus, prevent cancer cells from undergoing apoptosis. Interestingly, co-culture of fibroblasts with healthy cells drove a similar rewiring of the same signalling networks as to when oncogenic mutations, such as the classical G12D mutation in KRAS, were introduced in these healthy cells [70]. Moreover, several of the above-described therapies, including chemotherapy, depend on a functional immune system for effective tumour elimination. This effect is also known as immunogenic cell death (ICD), in which the cytotoxic agents-induced cell death of cancer cells leads to the recruitment of immune cells that enhance the efficacy of these drugs [71,72,73,74,75]. By modulating the immune landscape in the tumour microenvironment [68], fibroblasts can also affect the response of cancer cells to a multitude of agents including chemotherapies. As expected, the influence of fibroblasts on the immune subpopulations and immune infiltrate in the tumour is of major importance in determining the outcome of cancer cells to several immunotherapies. These processes are shown in Figure 1 and will be described in detail in the sections below.

#### 2.2.1. Cytotoxic Agents—Chemotherapy and Radiotherapy

An important way through which fibroblasts modulate the response of tumours to cytotoxic agents, such as chemotherapy and radiotherapy, is via their interaction with CSCs. These cells are at the top of the tumour hierarchy since, in addition to their self-renewal capacity, they are also able to differentiate into committed tumour cells and, consequently, repopulate the whole tumour [76]. CSCs have been shown to be intrinsically more resistant to these agents, not only due to an enhanced DNA damage repair mechanisms or a higher expression of ATP-binding cassette (ABC) transporters [77], but also due to their cycling state. CSCs are often in a dormant, non-proliferative state [78] that makes them relatively refractory to treatments that target an active proliferative state in cells. A mechanism through which fibroblasts have been shown to drive the expansion of the CSCs fraction in tumours was described by Su and colleagues [60]. The authors showed that a subpopulation of fibroblasts characterised by the expression of two markers, CD10 and GPR77, promoted chemoresistance by driving the survival of CSCs in both breast and lung cancer models. Interestingly, they showed that this subpopulation of fibroblasts is present in the tumours prior to any treatment, and that it becomes enriched upon therapy. CD10^+^GPR77^+^ fibroblasts were characterised by the activation of the NF-κB pathway, and their secretion of IL-6 as well as IL-8 was essential to drive resistance to chemotherapy. Based on more recent knowledge on the different subsets of fibroblasts, it could be hypothesised that these GPR77- and CD10-positive fibroblasts form a subcluster of the iCAFs. Kanzaki [79] has shown that, in breast cancer, the expression of CD10 and GPR77 is restricted to one of the clusters identified in a study from Bartoschek and colleagues [35]. Unfortunately, in this early study, no inflammatory CAF cluster was defined and the parallel is thus hard to establish. Nevertheless, the study from Su et al. further highlights the importance of uncovering fibroblast heterogeneity in the context of cancer. Moreover, it supports the significance of regarding the fibroblast dynamics during therapy. Several other studies have described mechanisms through which fibroblasts can support and expand tumour-initiating cells. Boelens et al. uncovered a paracrine crosstalk between cancer cells and stromal fibroblasts in which RIG-I and NOTCH signalling cooperate to drive the expansion of therapy resistant CSCs [59]. Briefly, they showed that exosomes secreted by stromal fibroblasts activate the anti-viral machinery in cancer cells via STAT1, which, in turn, drives the expression of interferon-stimulated genes (ISGs). This signature had previously been identified as a gene signature for radiotherapy and chemotherapy resistance [80,81] and was termed as interferon-related DNA damage resistance signature (IRDS). By facilitating the expression of NOTCH target genes in a STAT1-dependent manner, the stromal interaction with cancer cells drove the expansion of CSCs and, consequently, resistance to chemotherapy. Further studies have implicated JAK-STAT and NF-κB signalling in CAFs in the support of tumour-initiating cells. In pancreatic cancer, Chan et al. demonstrated that the secretion of ELR^+^ chemokines, CXCL1, CXCL2, CXCL5 and CXCL6 by CAFs bind and activate the CXCR2 downstream signalling in cancer cells, driving the expansion of CSCs [82]. Other pathways that modulate stem-like features in cancer cells have also been described to be activated by fibroblasts, namely Hedgehog [61] and Wnt signalling [83,84].

Several of the mechanisms described above seem to be mediated by iCAFs. Nevertheless, the myCAFs and ECM they synthesise can also strongly affect the way tumours respond to therapy. For example, a desmoplastic tumour can create a physical barrier that will prevent the exposure of cancer cells to the drug [85,86]. Moreover, the activation of integrin signalling in cancer cells as a result of a dense ECM, can promote the survival of malignant cells [87,88]. Modulation of the ECM by fibroblast-derived Anexin A6-loaded extracellular vesicles (EVs) also resulted in integrinβ1-FAK-YAP activation and drove chemoresistance in gastric cancer [89]. The secretion of immunosuppressive factors such as TGF-β by myCAFs can inhibit ICD and, consequently, impair the effect of radiation and chemotherapy in tumours [90]. In addition to its effect in immune cells, TGF-β can also modulate the processes of epithelial-to-mesenchymal transition (EMT) in cancer cells as well as regulate the expansion of CSCs [91].

#### 2.2.2. Targeted Therapies

In breast cancer, numerous targeted therapies have been used for a long time as the first line of treatment. These include endocrine therapies that modulate oestrogen receptor signalling [92] and, more recently, anti-HER2 therapies [93,94]. In endocrine-treated patients, a subset of fibroblasts characterised by the expression of CD146 were shown to regulate the response of cancer cells to the treatment. Briefly, CD146^-^ fibroblasts were able to activate IGF1R tyrosine kinase signalling in the cancer cells that drove their oestrogen-independent growth, and was, therefore, hypothesised to be the cause of the irresponsiveness of cancer cells to the treatment [95]. Moreover, CAF-secreted IL-6 acted in an autocrine fashion to drive the secretion of mir221/222-loaded microvesicles, which, in its turn, induced a stem-like phenotype in cancer cells [96]. In another hormone-driven tumour type, namely prostate cancer, Zhang et al. have shown that the secretion of NRG-1 by fibroblasts could result in resistance against anti-androgen therapy [97]. We have recently shown that CAFs in luminal breast cancer can be clustered based on their expression of NRG-1 [53]. It would be interesting to understand if stromal fibroblasts expressing high-NRG-1 could be involved in modulating the response of cancer cells to hormone therapy. CAFs might also play an important role in determining the success of tumour responses to anti-HER2 therapies such as trastuzumab, since the effectiveness of this therapeutic antibody is known to depend on the induction of an immune response, similarly to chemotherapy and radiotherapy [74,98,99]. Due to their immunosuppressive role, CAFs can antagonize the effect of trastuzumab [100]. Additionally, the activation of fibroblast growth factor (FGF) receptor 2 (FGFR2) by CAF-derived fibroblast growth factor 5 (FGF5) results in the activation of c-Src, which bypasses the inhibition of HER2 by lapatinib, a TKI used in the treatment of HER2-positive patients [101]. The induction of the PI3K/AKT/mTOR pathway in cancer cells by fibroblasts can also promote the survival of cancer cells and impair their response to anti-HER2 kinase targeted therapies [102].

In more recent years, the advances in cancer research have provided the basis for identifying novel oncogenic vulnerabilities and molecular drivers of disease. With these advances, novel targeted therapies, including several kinase inhibitors (KI) such as BRAF inhibitors (BRAFi), EGFR inhibitors (EGFRi) and CDK4/6 inhibitors, have been introduced in the clinics [103,104,105,106] for the treatment of a variety of cancer entities. It is now increasingly evident that stromal fibroblasts are also important in dictating the outcome of cancer cells to this type of therapy. In melanoma, BRAFi can lead to an enhanced ECM deposition by fibroblasts that induced high integrin β1-FAK signalling in cancer cells. This resulted in the activation of MAPK/ERK signalling, which helped overcome the inhibitory effect of the BRAFi [64]. Hepatocyte growth factor (HGF) production by stromal fibroblasts can also drive the activation of MAPK/ERK signalling in cancer cells, which can reduce the efficacy of targeted therapy [67,107].

#### 2.2.3. Immunotherapy

Another important component of the TME are the immune cells. Pioneering work by James Alisson [108] and Tasuko Honjo [109,110,111] in immune regulation has revolutionised the field of cancer immunology and established the groundwork for the development of numerous immunotherapies. Their respective findings that CTLA-4 and PD1 immune checkpoints inhibit the activity of cytotoxic T-cells and allow tumours to grow, led to the design of inhibitors against these molecules with the goal to enhance T-cell-mediated cell death of tumour cells [112]. These checkpoint inhibitors showed remarkable results, especially in melanoma patients [113,114,115]. However, it is not fully understood what factors dictate their efficiency and the durability of the patients’ response to these therapies. In addition to checkpoint inhibitors, several other immunotherapies are now available, including cancer vaccines [116,117] and the adoptive transfer of immune cells, such as chimeric antigen receptor (CAR) T-cells [118,119], and oncolytic viruses [120]. Due to the strong influence that fibroblasts can exert in the immune milieu of the TME [68,121,122], this cell type has emerged as a key modulator of the outcome of patients to these therapies. The aforementioned immunosuppressive molecules secreted by fibroblasts, namely TGF-β, IL-6, IL-8, CXCL12, can inhibit cytotoxic T-cell activity [123,124,125,126], drive myeloid-derived suppressor cell (MDSC) differentiation [127,128,129], modulate the phenotype of macrophages [130,131,132,133], promote Treg formation [54,134] and regulate the activity of natural killer (NK) cells [132,135]. In addition to these, further studies have described other mechanisms through which fibroblasts regulate the immune landscape in tumours. α-SMA+ CAF (myCAFs) also secrete metabolic reprogramming factors, such as IDO1, Arg2 and galectin, which are responsible for generating an immunosuppressive TME via inducing T cell anergy and inhibiting CD8^+^ T cell proliferation [136]. The production and deposition of ECM proteins by CAFs strongly regulate the T-cell location within the tumours [137]. A dense stroma can result in the exclusion of lymphocytes from areas rich in tumour cells, which results in poor adaptive immunity against the tumour [138,139]. Moreover, the production of certain ECM proteins by fibroblasts, namely tenascin C or thrombospondin 1 (TSP1), can negatively impact the adhesion of T-cells [140] and their activity [141] in the TME, respectively. Fibroblasts can express immune checkpoint molecules themselves, such as PD-L1 [142], PD-L2 [37,142] and B7H3 [37,143,144], which can all inhibit T-cell activation. The production of CCL5 by stromal fibroblasts leads to an immunosuppressive environment as a result of the recruitment of Treg cells into the TME [145]. The secretion of PGE_2_, which was recently shown to characterise the apCAF subpopulation, can also result in the expansion of regulatory T-cells [39]. Furthermore, PGE_2_ is capable of inhibiting NK cell function [146,147,148].

Fibroblasts not only communicate with the immune system via secreted factors, but they can also directly interact with CD8^+^ T-cells. The HLA-class I antigen-presentation by stromal fibroblasts along with the expression of PD-L2 and FASL results in the killing of antigen-specific cytotoxic T-cells [149]. Direct interaction between stromal fibroblasts and cancer cells also seems to drive resistance to oncolytic viruses as a result of the induction of a STING/IRF3-dependent inflammatory program in fibroblasts, which upregulates IFNβ1. The secretion of IFNβ1 into the TME induces an IFN-transcriptional program in cancer cells, rendering them less sensitive to infection by oncolytic viruses [150].

### 2.3. Impact of Anti-Cancer Therapies in Fibroblasts

Further evidence that supports the study and detailed investigation of fibroblast dynamics during therapy is that stromal populations are strongly modulated and influenced by a wide variety of treatments applied in the clinics, ranging from cytotoxic agents to targeted agents [151,152,153].

#### 2.3.1. Direct Effects on CAFs

Most cytotoxic agents, such as chemotherapy and radiotherapy, lead to extensive DNA damage, which can result in cell cycle arrest and, consequently, senescence [154,155]. Senescence is associated with the secretion of a vast array of cytokines, chemokines and growth factors, such as CCL2, IL-6, VEGF and TGF-β, which can aid cancer cell survival and, thus, contribute to therapeutic failure [155,156,157]. Chan et al. showed that high doses of chemotherapy induce a potent remodelling of fibroblasts by activation of the JAK-STAT1 and NF-κB pathways, which results in the upregulation and secretion of several chemokines that, in their turn, support tumour-initiating cells and lead to chemoresistance [82]. Similarly, another study described an increased risk of developing resistance to chemotherapy after treatment with the maximum tolerated dose due to drug-induced changes in the tumour stroma [158]. In both cases, the effects observed in the tumour stroma could be prevented by adopting a metronomic (low-dose) chemotherapy regimen [82,158]. In general, chemotherapy seems to activate an inflammatory gene signature in stromal fibroblasts, which is associated with a pro-tumorigenic state [159]. The activation of the NF-κB signalling pathway by therapy-induced damage also promoted chemoresistance by driving the expression and secretion of WNT16B by fibroblasts and the subsequent activation of the Wnt program in cancer cells [160]. The secretion of interleukins (ILs) by CAFs has been reported after treatment with chemotherapy. For example, IL-17A derived from chemotherapy-treated CAFs led to the expansion of tumour-initiating cells and, consequently, was shown to contribute to therapeutic failure [63]. The exposure of fibroblasts to chemotherapy also resulted in a higher secretion of IL-11, which activated STAT3 signalling in cancer cells and drove the upregulation of anti-apoptotic pathways [161]. In addition to the effects mentioned above, chemotherapeutic agents can also drive a myCAF phenotype in stromal fibroblasts. Exposure to these drugs can increase the amount of secreted ECM by CAFs that, among other effects, can form a physical barrier and prevent the drug from reaching the cancer cells [85,162]. Chemotherapy-induced secretion of exosomes by fibroblasts can also play an important role in resistance to chemotherapy. In PDAC, Richards et al. showed that fibroblast-derived exosomes regulate EMT pathways in cancer cells in a Snail-dependent way and that targeting this pathway increased the sensitivity of cancer cells to chemotherapy [163]. Transfer of miRNAs from CAFs to cancer via secreted extracellular vesicles has also been shown to be induced by chemotherapy and to promote resistance [164].

Similar effects have been described for fibroblasts exposed to radiotherapy [151]. Strong desmoplastic reactions often characterise the irradiated areas. This is a consequence of fibroblast activation and enhanced deposition of ECM [165]. Indeed, an enrichment in αSMA-positive fibroblasts can be found in chemo and radiotherapy treated tumours [158,166] and stromal signatures are often associated with a worse outcome regarding disease free survival [166,167,168,169]. High amounts of ECM can then activate pro-survival pathways in cancer cells and impair the response to radiotherapy as well as to chemotherapy, in an integrin β1-dependent manner [87,88,170,171,172]. As with cytotoxic agents, radiation can drive a senescent phenotype in stromal fibroblasts, which is characterised by the expression of, among others, TGF-β [154]. The presence of elevated levels of TGF-β can then not only increase ECM production but also drive an immunosuppressive environment [154]. Tommelein et al. also found that irradiated fibroblasts could promote the survival of cancer cells via insulin-like growth factor receptor-1 (IGF1R) signalling. This pathway activation was driven by a senescence-like phenotype of the fibroblasts that was characterized by the secretion of IGF1 after treatment [173].

Despite their intended specificity, targeted therapies can also directly affect and modulate the phenotype of stromal fibroblasts. In colorectal cancer, the treatment of tumours with an EGFR inhibitor led to an increased secretion of EGF by stromal fibroblasts and conferred resistant of neighbouring cancer cells to the treatment via activation of MAPK signalling [174]. Matrix remodelling by fibroblasts exposed to BRAF inhibitors can also impair the response of cancer cells to the treatment. In this system, resistance was driven by elevated activation of integrin β1-FAK-Src signalling in malignant cells because of high ECM-production by fibroblasts [64].

#### 2.3.2. Indirect Effects in CAFs

Fibroblasts can further be indirectly modulated by therapy, since it has been described that cancer cells acquire a specific secretory profile after exposure to drugs, named therapy-induced secretome [175]. In the same way that cytotoxic agents can drive the senescence of fibroblasts, the cancer cells exposed to these agents can also undergo therapy-induced senescence and acquire the aforementioned senescence-associated secretory phenotype (SASP) [175], which then modulates the phenotype of fibroblasts. We have recently shown that cancer cells exposed to high doses of chemotherapy upregulate the expression of IFNβ1, which acted in a paracrine manner to drive a pro-tumorigenic state of fibroblasts that then drove the recovery of cancer cells after treatment [176]. This upregulation of IFNβ1 after treatment with cytotoxic agents goes in line with previous studies that have shown that high levels of damage in cancer cells after treatment results in the activation of the STING/IRF3 pathway and drives IFNβ1 expression [177,178,179]. Moreover, therapies such as chemotherapy and radiation can strongly modulate the immune milieu of the TME and, consequently, affect the profile of stromal fibroblasts [74]. Radiation can result in vascular damage, which triggers an inflammatory response and, consequently, promotes myofibroblast differentiation [151]. This vascular damage can also lead to hypoxia and increase the production of HIF1α [180,181]. In colorectal cancer, it was shown that HIFα and TGFβ cooperate to induce hedgehog transcription factor GLI2 expression in tumour-initiating cancer cells, which drives stemness and chemoresistance [62]. The increased secretion of TGF-β by melanoma cells after exposure to Vemurafenib, a BRAF-inhibitor, was also able to drive fibroblast activation. Fibroblasts were then shown to produce increased levels of ECM, but also growth factors such as NRG-1 and HGF. Combined, these factors promoted the survival of cancer cells to treatment [182]. Apicella et al. described a non-autonomous cancer cell mechanism in non-small cell lung cancer (NSCLC) that drove resistance to targeted therapies. Interestingly, they show that in vivo resistant cancer cells become re-sensitized to the therapy when treated in vitro in the absence of the TME. Mechanistically, they show that cancer cells treated with TKIs targeting MET and EGFR secrete higher levels of lactate, which then instructs CAFs to secrete HGF, resulting in the non-responsiveness of cancer cells to the treatment [58].

The effects of therapies in fibroblasts are vast and complex. Despite the attempts to understand these mechanisms, further investigations are required. All the studies thus far have addressed these alterations in a bulk-fashion and detailed single-cell studies are still lacking. Moreover, it is necessary to unravel whether the mechanisms described above are tumour type- or context-specific, to develop strategies to identify which patients would benefit from their inhibition and, importantly, at what time-point during treatment. Understanding these complex dynamics is hard, but it would provide essential insights for the targeting of this cell type for cancer treatment.

### 2.4. Targeting Fibroblasts in Cancer Treatment

Several strategies have been developed for the targeting of fibroblasts in cancer treatment. In pre-clinical models, attempts to (1) directly target fibroblasts, (2) target secreted molecules such as ECM or soluble signalling molecules or (3) inhibit pro-tumorigenic signalling pathways have been described. Regardless of the strategy used the major goals have been to either eliminate stromal fibroblasts in general or certain subpopulations from the tumour, or to normalise the stroma/tumour-stroma interactions to ensure that the crosstalk between fibroblasts and cancer cells or the immune milieu does not support tumour progression, invasion or therapeutic failure. A summary of these strategies is described in Figure 2.

#### 2.4.1. Direct Targeting of Fibroblasts

Numerous attempts of targeting fibroblasts have focused on their direct targeting via specific cell surface markers. Some early studies in PDAC have targeted αSMA-expressing fibroblasts. However, contrary to what had been expected, tumours in which this stromal population was eliminated grew at even higher rates, demonstrating that, at least in this tumour type, αSMA-positive fibroblasts restricted tumour growth [28,29]. Similar attempts to eliminate or directly target fibroblasts in the TME have instead focused on the FAP expressing fibroblast population. In this case, regression of the tumour was indeed observed [123,129]. Diverse FAP-targeting strategies have since been developed and tested in pre-clinical models, including genetic deletion [123,129], molecular inhibitors that block FAP enzymatic activity [129,183,184,185,186], anti-FAP monoclonal antibodies [187] and FAP-antigen vaccination [188], all with promising tumour-restraining results. Most of these studies correlate the observed effects with the impact of the treatment in the immune milieu. The tumours in which FAP was targeted were characterised by a reversion of the immunosuppressive environment and an increase in T-cell infiltration [123,185,186]. Consequently, clinical trials targeting this protein have been undertaken to investigate the potential of such treatments in patients. The usage of blocking antibodies, namely sibrotuzumab (a humanised anti-FAP antibody, F19), proved to be safe in a phase I trial [189,190], but failed to significantly improve overall survival. Similar results were obtained with the FAP-inhibitor PT100 (talabostat) [191,192,193,194]. Even though the application of anti-FAP antibodies showed very limited clinical efficacy, these agents exhibited very good stroma-targeting properties [190]. This has led to the development of antibody conjugates, in which FAP is used for the localised delivery of the conjugate. One such fusion antibody that is currently in a phase I clinical trial (NCT02627274) is called RO6874281, in which FAP is cross-linked with IL-2. The main goal of using this antibody is to activate T-cells in the TME [195]. Other strategies involve the conjugation of FAP with other immune modulators, such as IL15 [196], co-stimulatory ligands as B7.2 [197] and CD40 [198], or immunotoxins [199,200]. Moreover, FAP antibodies can be conjugated with agents that will directly induce apoptosis, such as the cytotoxic drug DM1, which have shown potent inhibitory activity in pre-clinical models [187]. Finally, the ablation of FAP-expressing cells in the TME was also achieved by using CAR-T cells directed against FAP [201,202].

In summary, a broad variety of strategies targeting FAP are available, although clinical trial results have been disappointing thus far. This might be improved with optimised strategies to directly target and eliminate FAP-positive cells, such as conjugated antibodies, vaccination or bispecific CAR-T cells. The enhanced immune response that has been observed in the pre-clinical models suggests that a combination of these strategies with other therapies, such as immune checkpoint inhibitors or chemotherapy, will likely be advantageous. Moreover, several studies clearly showed that these approaches are highly effective to target specific conjugates to the tumour site, which can be further explored.

Another cell surface marker that has been explored in pre-clinical models is GPR77. This protein, in combination with CD10, has been shown to identify a subpopulation of fibroblasts that is responsible for driving chemoresistance. The blockage of GPR77 using a neutralising antibody was shown to reduce the CSCs in tumours and enhance the response to chemotherapy [60]. Moreover, since these fibroblasts were identified as being present in the tumour prior to treatment, the concomitant targeting of these with the chemotherapeutic regiments could likely provide a benefit for the patient. Further studies along these lines are required.

#### 2.4.2. Targeting of the ECM

Stromal rich tumours can have impaired drug delivery as a result of the physical barrier presented by the ECM. Moreover, the ECM can activate pro-survival signalling pathways in tumour cells. This makes approaches targeting the ECM and/or its downstream signalling attractive for the treatment of a variety of tumours. Numerous pre-clinical studies blocking integrin signalling have shown that this axis is critical for the development of drug resistance and that its abrogation in combination with cytotoxic agents could improve therapy response and overall survival [87,203]. A blocking antibody, FG-3019, that interferes with integrin signalling activation is being tested in a phase I/II clinical trial (NCT02210559). Moreover, the biosafety of defactinib, a small molecule inhibitor that targets FAK and thereby prevents downstream pathway activation in cancer cells, has been investigated in a phase I clinical trial (NCT02546531) [204]. Direct targeting of ECM proteins, namely fibronectin [205], tenascin C [206] and hyaluronan [138,207,208] have also been studied in pre-clinical models with promising results. The targeting of hyaluronan for degradation using a PEGylated enzyme, PEGPH20, in combination with chemotherapy showed positive results in early phase clinical trial in patients with PDAC [209,210]. Losartan, a small molecule inhibitor that targets the angiotensin receptor and, consequently, leads to a decrease in hyaluronan levels, is currently tested in a phase I clinical trial (NCT03563248) [211]. Another strategy employed in pre-clinical trials exploits the inhibition of matrix metalloproteinases (MMPs). These enzymes are critical players in the remodelling of the ECM [212]. Unfortunately, disappointing results have emerged from all the MMP inhibitors tested in clinical trials thus far [213]. Novel strategies are also emerging, with the use of CAR-T cells engineered to express heparanase, an enzyme that degrades ECM proteins, having shown promising pre-clinical results by enhancing T cell infiltration and anti-tumour activity [214].

#### 2.4.3. Targeting of Cytokines and Growth Factors

One of the most extensively studied and described pro-tumorigenic axes involved in the communication between cancer cells and their microenvironment is the IL-6/IL-6R/JAK-STAT3 pathway. Not surprisingly, attempts to target this pathway soon emerged and were tested in pre-clinical models [215]. The targeting of this axis is threefold, since (1) it has been described to be involved in the activation of fibroblasts [216,217], (2) its effect in tumour cells drives pro-tumorigenic states by, among others, modulating stemness and invasiveness [218] and (3) it negatively regulates tumour-infiltrating immune cells [219]. An IL-6 monoclonal antibody, siltuximab, was tested in a phase II clinical trial for the treatment of prostate cancer and demonstrated a safe profile but failed to improve patient outcome [220]. The efficacy of this antibody is currently evaluated in a phase Ib/II trial for the treatment of advanced pancreatic cancer (NCT04191421). Furthermore, several clinical trials investigating the activity of molecules that inhibit downstream signalling of IL6-, namely ROCK and STAT3 inhibitors, have been undertaken. A dual ROCK-AKT inhibitor, AT13148, was tested in phase I, but failed to show a safe profile [221]. The STAT3 inhibitor, AZD9150, was well tolerated in a phase I clinical trial [222] and an investigation of its efficacy is undergoing.

Another major pro-tumorigenic molecule that is secreted by CAFs is CXCL12. This chemokine is mostly involved in the modulation of immune cells in the TME, and its targeting can alleviate immunosuppression and drive effective anti-tumour immunity. Therefore, the inhibition of this axis using antagonists or antibodies targeting its receptor, CXCR4, has been extensively studied [223,224]. The efficacy of some of these molecules, namely the CXCR4 antagonist AMD3100, is currently evaluated in clinical trials for the treatment of several tumour entities, including haematological malignancies and advanced pancreatic cancer (https://www.cancer.gov/about-cancer/treatment/clinical-trials/intervention/plerixafor, accessed on 1 June 2021), after having displayed a safe profile in phase I clinical trials [225,226].

The inhibition of Shh and TGFβ has also been extensively studied in an attempt to ‘normalise’ the tumour stroma by reverting the activated state of fibroblasts. Moreover, these molecules have broad activities and are capable of regulating several other processes, as described in the previous sections. Despite the exciting results of Shh inhibition in pre-clinical studies [52,61,227], evaluation of the FDA-approved inhibitors, saridegib and vismodegib, in early phase clinical trials showed very disappointing results with no improvement in disease-free survival (DFS) and overall survival (OS) (NCT01064622; NCT01088815) [228]. To date, several TGFβ inhibitors have been developed and their efficacy studied using pre-clinical models [229,230,231]. Following these results, clinical trials have been established and, while some have failed to demonstrate a significant overall benefit, other trials showed a benefit of inhibiting TGF-β signalling [231].

Despite the many attempts to target the TME and to significantly improve patient outcomes, most of these strategies have fallen short. Improved combinatorial strategies will likely be required to achieve an effective inhibition of tumour growth. Moreover, most of the clinical trials enrol patients with heavily pre-treated tumours and advanced/metastatic disease, which can hinder the effectiveness of these drugs. The rational targeting of these axes in early treatment could provide an advantage by preventing the development of resistance and inhibiting the invasive properties in cancer cells. However, more studies are required to evaluate this hypothesis. Furthermore, increasing knowledge in CAF biology and the unravelling of subpopulation-specific markers will likely drive the development of more effective treatments as this might allow the targeting of particularly relevant subpopulations with enhanced specificity. Finally, the identification of biomarkers and the better stratification of patients will be necessary to see a potential improvement in disease-free survival and overall survival in particular patient groups, as with other targeted therapies in cancer therapy.

## 3. Conclusions

The role of stromal fibroblasts in the TME is complex. Advances in molecular biology have just started to unravel how heterogeneous the function of these cells are. Several subpopulations with distinct roles have been identified and further dissection will likely drive the discovery of further relevant markers and states of this cell type. Another layer of complexity arises when the impact of the numerous treatments in these cells is brought into the picture. The drugs used in the treatment of cancer strongly modulate both the phenotype and the secretory profile of fibroblasts, which should be investigated in detail in the future. A broad comprehension of the interactions between stromal fibroblasts and the surrounding cells will be of the utmost importance toward the development of novel targeted strategies that can improve the outcome of cancer patients.

## Figures and Tables

**Figure 1 cancers-13-03526-f001:**
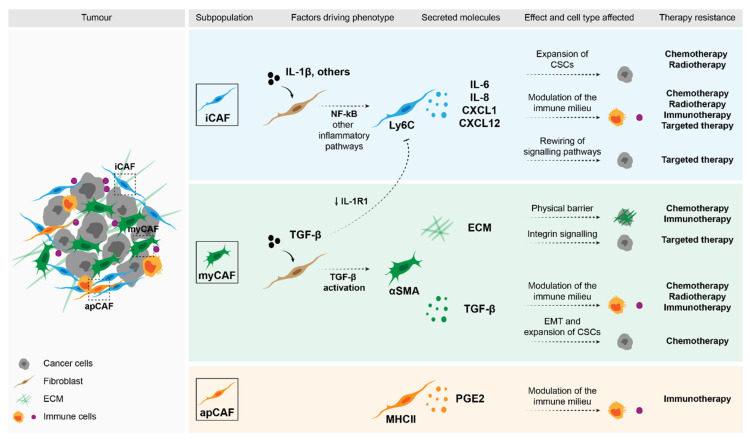
Impact of fibroblasts in therapy response. Stromal fibroblasts can drive therapy resistance through a variety of mechanisms. Factors that were shown to be secreted by specific subpopulations of CAFs can directly affect malignant cells and drive the expansion of CSCs (e.g., cytokines such as IL-6, IL-8 and CXCL1), or activate pro-survival pathways (e.g., ECM through integrin signalling). Moreover, these secreted molecules can modulate the immune populations in the TME and in this way, determine therapy response.

**Figure 2 cancers-13-03526-f002:**
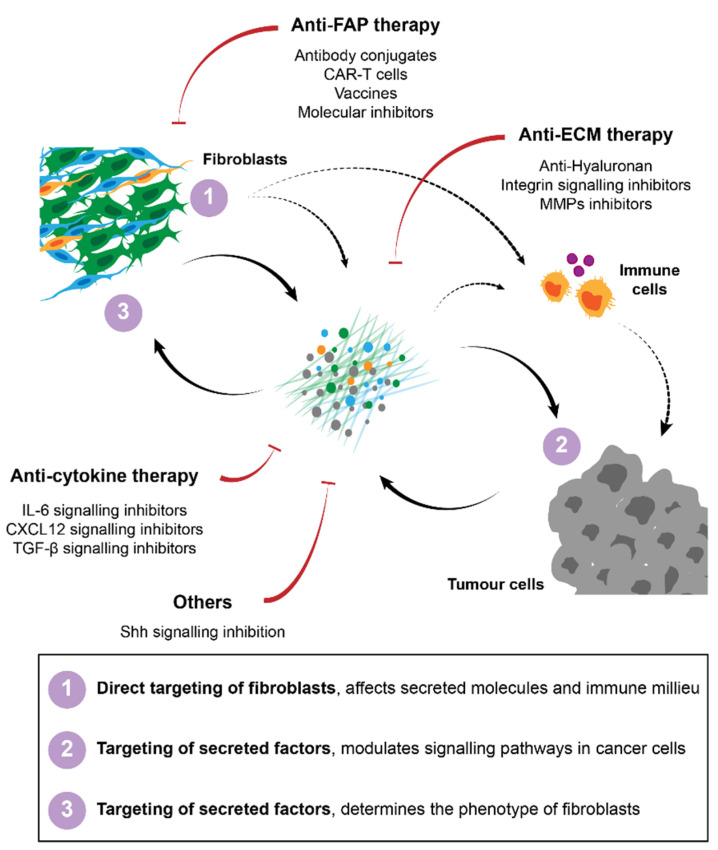
Strategies for cancer therapy targeting fibroblasts and their interactions in the TME. Several approaches have been proposed to target CAFs, including direct targeting and elimination of specific subpopulations of fibroblasts, namely FAP-positive cells. Therapies against secreted factors and their respective signalling, such as ECM or cytokine signalling inhibitors have also been developed to improve immunity and to block pro-tumorigenic interactions between CAFs and their surroundings.

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
