# Peer review of "Cancer-Associated Fibroblasts: Implications for Cancer Therapy"

_cancers, 2021, doi:10.3390/cancers13143526_

Round 1
Reviewer 1 Report
This is an excellent and comprehensive review, and just requires some minor changes, mostly to address some minor errors or overly complicated English. These have been highlighted in the attached annotated PDF.

Author Response
We would like to thank the reviewer for thorough and critical reading of our manuscript as well as for her/his very positive comments on the work.
The highlighted changes in the manuscript have been addressed and corrected as requested by the reviewer.
Reviewer 2 Report
In the review article "CANCER-ASSOCIATED FIBROBLASTS: IMPLICATIONS FOR CANCER THERAPY," the authors provide a very well-organized review of the influence and potential benefits of tumor-associated fibroblasts in tumor progression and therapy. All important aspects are addressed and discussed. I therefore recommend that this work be accepted for publication in its current form.
Author Response
We would like to thank the reviewer for thorough and critical reading of our manuscript as well as for her/his very positive comments on the work.
Reviewer 3 Report
This is a good review article and can be accepted for publish in Cancers with minor revisions: (1) The authors can add more their own insights instead only summarizing cancer-associated fibroblasts (2) Layout design can be better. For example, the quality of figures can be improved by professional drawing, which can contribute the citations thereafter.
Author Response
We would like to thank the reviewer for thorough and critical reading of our manuscript as well as for her/his very positive comments on the work.
We thank the reviewer for her/his comments. There are several points in the manuscript where we have provided our own insights. Please see lines 232-235, 347-349, 354-355, 396-399, 556-563, 690-703. More importantly, we integrated recent knowledge on fibroblast heterogeneity with previously described mechanisms through which fibroblasts drive resistance (this can be seen in figure 1 and in section 2.2.). Regarding the figures we do not agree that these need to be changed. We believe that they are clear and comprehensive and they were provided in high quality (300 dpi).